# The Threat of Micro-/Nanoplastics to Male Fertility: A Review of the Data and the Importance of Future Research

**DOI:** 10.3390/ijms262311457

**Published:** 2025-11-26

**Authors:** Shawn Aji Alex, Nevin K. George, John Guardiola, Deborah Clegg

**Affiliations:** Paul L. Foster School of Medicine, Texas Tech University Health Science Center at El Paso, El Paso, TX 79905, USA; shawn.alex@ttuhsc.edu (S.A.A.);

**Keywords:** microplastics, nanoplastics, endocrine-disrupting chemicals, testicular function, germ cells, Leydig cells, Sertoli cells

## Abstract

Micro-/nanoplastics (MNPs) and their associated endocrine-disrupting chemicals (EDCs) have emerged as pervasive environmental pollutants, with growing concern for their impact on male reproductive health. In this review, we synthesize the findings from twenty-one peer-reviewed studies published between January 2019 and March 2025, selected through a structured literature search conducted in accordance with the SANRA guidelines. Emphasis was placed on studies examining the cellular effects of MNPs and EDCs on Germ, Leydig, and Sertoli cells. The literature indicates multiple mechanisms of testicular toxicity, including degradation of the blood testis barrier, disruption of signaling pathways critical for spermatogenesis and hormone synthesis, induction of oxidative stress and inflammation, and structural and genetic damage to testicular tissues. These data, primarily derived from in vitro and animal models, not only highlight significant biological disruptions but also underscore the limitations in extrapolating results to human physiology. Differences in exposure routes, dosages, and species-specific responses present challenges to direct translation to humans. This review concludes that further human-centric research that mimics real-life exposure and impacts is essential to assess chronic, low-dose exposures and bridge the gap between experimental data and real-world reproductive outcomes, ultimately informing public health strategies and guiding future investigations into the reproductive risks posed by MNPs and EDCs.

## 1. Introduction

Plastic invention has provided many benefits to society and enabled advancements in diverse fields, such as consumer product packaging, home appliances, medical equipment, electronics, clothing, and construction. The current plastic production is estimated to exceed 450 million metric tons annually, due to its versatile nature and low manufacturing cost [1]. However, the widespread use of plastics has also contributed to significant negative impacts on human health, the environment, and the global economy, and plastic pollution has become a severe threat to the planet and a growing concern. It is expected that 11 billion tons of plastics will be present in the environment by 2025 [2]. One of the most pressing issues arising from the ubiquity of plastics is the emergence of microplastics and nanoplastics (MNPs)—tiny plastic particles defined as 100 nm to 5 mm and less than 100 nm, respectively. These particles are further categorized as primary or secondary MNPs based on their origin [3]. Primary MNPs are intentionally manufactured for use in consumable products, such as cosmetics, scrubs, body wash, and toothpaste, while secondary MNPs are generated through the breakdown of discarded plastic products via long-term UV exposure or biodegradation, ultimately fragmenting into microscopic pieces [4]. MNPs may be lipophilic or hydrophobic and appear in various shapes and sizes, influencing their transportation, distribution, and absorption into human tissues [5]. Contaminants such as endocrine-disrupting chemicals (EDCs) are often incorporated into microplastics to alter their properties and functions [6]. Plastics have infiltrated a wide range of environments due to their pervasive use, including oceans, freshwater systems, sediments, inland soils, air, and food products (e.g., drinking water, seafood, and crops) [7]. The prevalence of MNPs in food, air, consumer products, and the environment has led to increased human exposure through ingestion, inhalation, and dermal contact [8]. MNP exposure not only directly threatens human health but also poses an additional risk through biomagnification—a process in which the contaminant concentration increases within an organism relative to the levels present in its prey—because humans are at the top of the food chain [9]. These pervasive environmental contaminants have raised serious concerns regarding their potential risks and long-term impacts on human health. 

In this manuscript, we highlight multiple mechanisms through which MNPs may impair testicular function, including disruption of the blood testis barrier, interference with signaling pathways essential for spermatogenesis and hormone synthesis, induction of oxidative stress and inflammation, and structural and genetic damage to testicular cells. These particles may also act as carriers for EDCs, with co-exposure amplifying their reproductive toxicity. Overall, the data generated in these studies underscore an urgent need for greater awareness and targeted interventions to mitigate the impact of plastic pollution on male reproductive health.

## 2. Results

Although the effects of MNPs are still being investigated, studies have shown that they significantly impact multiple systems in the body, including the digestive, respiratory, cardiovascular, immune, reproductive, and renal systems [10]. Once inside the body, MNPs can induce inflammatory responses, oxidative stress, and tissue fibrosis, and they may act as vectors for pathogens and toxic substances [10]. 

Research has highlighted that exposure to environmental pollutants—including EDCs, heavy metals, and MNPs—can adversely affect reproductive health, and these effects are particularly pronounced in the testes, where critical developmental processes such as spermatogenesis occur, potentially leading to long-term reproductive consequences [11]. Because human germ cell maturation and other key fertility-related events involve tightly regulated steps, the reproductive organs are especially vulnerable to environmental toxins, which can disrupt these processes and result in subfertility [12]. Subfertility is defined as any form of reduced or complete loss of fertility after 12 months or more of unprotected sexual intercourse, with males accounting for 20–30% of all cases [13]. The global prevalence of subfertility is estimated to affect approximately 60–80 million couples, representing about 8.1% of all couples worldwide [13]. Male subfertility has increased dramatically in many developed countries, with previous studies noting increases in inadequate sperm counts and poor semen quality as contributing factors [14]. 

In this review, we highlight the recent studies on the impact of MNPs and their associated EDCs on the human male testes, focusing on findings from various in vitro and in vivo studies and extending the investigation to assess the knowledge gaps and broader MNP/EDC effects on male reproductive health.

### 2.1. Role of Testes in Fertility

Male reproductive function is delicately regulated by the coordinated activity of the hypothalamic pituitary testes (HPT) axis. Within this axis, the hypothalamus secretes gonadotropin-releasing hormone (GnRH), which stimulates the pituitary gland to produce follicle-stimulating hormone (FSH) and luteinizing hormone (LH), which exert their reproductive effects on the testes by binding to their respective receptors [15]. The testes play a central role in male fertility through both sperm production and androgenic hormone synthesis. Spermatogenesis—the process of developing new sperm cells—occurs within the seminiferous tubules, where Sertoli cells support germ cell development and inhibit immune cell attack and apoptosis, a phenomenon known as immune privilege [16]. This process is supported by testosterone production from Leydig cells and FSH/LH stimulation from the pituitary gland [16]. Androgenic hormones produced in the testes are essential for the development and maintenance of reproductive functions, including the regulation of spermatogenesis, sexual functioning, and Sertoli cell activity [17]. These hormones—primarily testosterone and its precursors—also play a critical role in the differentiation of external genitalia during fetal development and in the maintenance of sex-specific external characteristics. Furthermore, they are pivotal in the feedback regulation and interplay with hypothalamic pituitary hormones such as GnRH, LH, and FSH [18], and MNPs and their associated EDCs have been shown to adversely affect this delicate balance. The disruption mechanisms include disorganization of the HPT axis through feedback interference, degeneration of germ cells (spermatogonia, spermatocytes, and spermatids), and damage to Sertoli and Leydig cells. Additional effects include the reduced antioxidant activity, activation of oxidative stress pathways, and induction of apoptosis—all of which will be discussed in the subsequent sections.

### 2.2. Germ Cells (Spermatogonia, Spermatocytes, Spermatids, and Sperm)

The seminiferous tubules are highly protected and tightly regulated regions of sperm production. However, MNPs interfere with the blood–testis barrier, and disturbances in this barrier are associated with the disruption of the spermatogenic process [19]. In an in vitro study of human spermatozoa, exposed to 30 min of polystyrene at concentrations of 0.1, 1, and 10 µg/mL, MNPs led to acrosomal and plasma membrane damage, DNA fragmentation, and reactive oxygen species (ROS)-induced injury [20]. This study further confirmed sperm DNA damage by demonstrating the increased expression of heat shock protein-70 (HSP70), a molecular chaperone and protective factor against sperm damage, following microplastic exposure [20,21]. Another study found that MNPs can also induce DNA damage in mouse sperm via the phosphoinositide 3-kinase (PI3K)/protein kinase B (Akt) signaling pathway [22]. 

Parallel studies in male mice have linked the disruption of the sperm acrosome structure and biogenesis to the downregulation of key marker genes (Gba2, Pick1, Gopc, Hrb, Zpbp1, Spaca1, and Dpy19l2), all of which are essential for acrosome formation [23]. These findings were supported by ultrastructural alterations in the testes. The same study also associated MNP exposure with autophagy repression and the failure to correct acrosomal defects due to the downregulation of Gopc and Dpy19l2 [23]. 

Beyond structural damage, MNPs have also been implicated in the impairment of testicular spermatogenesis, leading to reduced sperm cell quality. This toxicity was observed in vitro in mouse spermatocytes, where increased levels of inflammatory markers—interleukin IL-1β and IL-6—following MNP exposure activated the nuclear factor erythroid 2-related factor 2 (Nrf2)/heme oxygenase-1 (HO-1)/nuclear factor-kappa B (NF-kB) signaling pathway, resulting in oxidative stress and testicular injury [24]. In addition to NF-kB activation, MNPs can induce inflammation, abnormal spermatogenesis, and compromised sperm quality through ROS-mediated signaling in mouse sperm cells [24,25]. MNP exposure also upregulates oxidative stress via the activation of the JNK and p38 MAPK pathways, triggering pro-inflammatory cytokine release in mice [26]. Another study conducted on the sperm cells of mice reported elevated malondialdehyde levels and altered activities of ROS-scavenging enzymes—superoxide dismutase and catalase—after maternal and postnatal MNP exposure, contributing to testicular oxidative injury [27]. Collectively, these findings demonstrate that MP exposure disrupts sperm quality and function through oxidative damage.

Additional studies on mouse sperm cells have shown that MNP exposure reduces sperm capacitation—the process that prepares sperm for fertilization—by diminishing F-actin polymerization [28]. This reduction is mediated by the increased ubiquitination of Rho GTPases, including Ras-related C3 botulinum toxin substrate 1 (RAC1) and cell division cycle 42 (CDC42) [28]. Other studies on mouse and rat models have confirmed declines in sperm motility and quality through the reduced activity of key metabolic enzymes, such as lactate dehydrogenase (LDH), succinate dehydrogenase (SDH), and B-cell lymphoma 2 (Bcl-2), alongside elevated malondialdehyde (MDA) and ROS levels [26,29]. Collectively, these studies, although conducted on animal models, provide direction by suggesting that MNP exposure contributes to subfertility through testicular structural and DNA damage, impaired sperm architecture, increased ROS generation, apoptosis, and oxidative stress (see Table 1 and Figure 1 and Figure 2).

### 2.3. Leydig Cells

As previously mentioned, MNPs disrupt the blood–testis barrier and accumulate within testicular tissue, leading to inflammation and impaired Leydig cell function in mouse models [28]. MNP exposure has been shown to cause histological changes, including a reduced Leydig cellular area and decreased sensitivity to Insulin-Like Factor 3 (INSL3)—an essential ligand involved in multiple male sexual functions—resulting in Leydig cell degeneration [30]. 

Studies have also demonstrated that MNPs downregulate luteinizing hormone (LH)-mediated signaling through the LH receptor (LHR)/cyclic adenosine monophosphate (cAMP)/protein kinase A (PKA)/steroidogenic acute regulatory protein (StAR) pathway, which is critical for testosterone production in Leydig cell cultures from chronically exposed mice [31,32,33]. This downregulation is driven by the activation of pro-inflammatory M1 macrophages, TNF-α upregulation, and nuclear factor-κB (NF-kB) signaling, leading to the transcriptional repression of the LHR/cAMP/PKA/StAR axis [31]. Another study conducted on mouse models found that the transcriptional suppression of the LHR/cAMP/PKA/StAR pathway was associated with the upregulation of hypoxia-inducible factor 1-alpha (HIF-1α), mediated through the activation of the mitochondrial ERK1/2 MAPK/mTOR and AKT/mTOR signaling pathways [33]. Similarly, MNP exposure led to diminished testosterone production due to plasma membrane damage, mitochondrial dysfunction, and increased apoptosis in Leydig cells [34]. These findings suggest that there are multiple mechanisms that contribute to reduced testosterone synthesis and Leydig cell viability following MNP exposure. In parallel, MNP exposure has been linked to oxidative stress through the suppression of GPX1, a key antioxidant enzyme, and the upregulation of the PERK–EIF2α–ATF4–CHOP axis and other endoplasmic reticulum (ER) stress pathways in mouse models [35]. Another study reported that MNPs induce oxidative stress by impairing antioxidant defenses in Leydig cells via mitochondrial-associated ER membrane (MAM) dysregulation, leading to increased autophagy and apoptosis in mouse Leydig cells [36]. Elevated oxidative damage further contributes to decreased serum GnRH, LH, FSH, and testosterone levels, ultimately also impairing Leydig cell function in mouse studies [28,35]. These findings suggest that MNP exposure disrupts the Leydig cell integrity through both structural alterations and functional impairment, resulting in reduced testosterone levels and compromised male reproductive function (see Table 1b) in various mouse Leydig cells.

**Table 1 ijms-26-11457-t001:** (a) Summary of the potential adverse effects of MNPs on germ cells. These studies suggest MNP exposure contributes to subfertility through testicular structural and DNA damage, impaired sperm architecture, increased ROS generation, apoptosis, and oxidative stress in model systems, but needs to be validated and verified if these same processes occur in humans. (b) Summary of the potential adverse effects of MNPs on the Leydig cells. These data collectively suggest that MNP exposure disrupts Leydig cell integrity through both structural alterations and functional impairment, resulting in reduced testosterone levels and compromised male reproductive function in model systems. (c) Summary of the adverse effects of MNPs on the Sertoli cells. These data suggest MNPs pose a risk to the structural and functional integrity of Sertoli cells, with potential consequences for male fertility in model systems.

(**a**)					
Germ cells	Source	Animal/Model	Target Cells	Finding	Mechanism
	[20]	Human	Spermatozoa	Sperm acrosomal and plasma membrane damage	DNA fragmentation and reactive oxygen species (ROS) damage leading to elevated HSP70 expression
	[23]	Mouse	Testicular spermatocytes germ cells	Sperm acrosome structure and biogenesis disruption	Modification of the autophagy and ultrastructure of acrosome and downregulation of acrosomal formation
	[22]	Mouse	Sperm	Sperm DNA damage	Increased PI3K/Akt signaling pathway
	[24]	Mouse	Testicular spermatocytes germ cells	Sperm quality decline	Elevated NF-κB/Nrf2/HO-1 signaling pathway leading to inflammation and abnormal spermatogenesis
	[26]	Mouse	Sperm	Increased oxidative damage	Elevated p38 MAPK and JNK signaling pathway and reduced normal sperm metabolism enzymes
	[27]	Mouse	Sperm	Trans-generational spermatogenesis disruption	Increased oxidative testicular injury leading to disrupted seminiferous epithelium and decreased sperm count
	[28]	Mouse	Sperm	Decreased sperm quality and count and testicular microstructure and function alteration	Elevated ubiquitination levels of sperm RAC1 and CDC42 leading to sperm capacitation inhibition. Increased expression of genes for apoptosis and inflammation
	[29]	Rat	Sperm germ cells	Increased ROS generation and apoptosis	Raised levels of ROS and MDA
(**b**)					
Leydig cells	Source	Animal/Model	Target Cells	Finding	Mechanism
	[30]	Mouse	Testis	Reduced Sertoli cell number, Leydig cell area, and differentiation	Reduced sensitivity to INSL3 leading to Leydig cell degeneration
	[31]	Mouse	Leydig cells	Reduced testosterone production	LH-mediated LHR/cAMP/PKA/StAR pathway downregulation
	[33]	Mouse	Leydig cells	Leydig cell membrane damage	ERK1/2 MAPK/mTOR and AKT/mTOR signaling pathways activation
	[34]	Mouse	Leydig cells	Mitochondrial impairment, apoptosis, and reduced testosterone production.	Increased ROS burst, oxidative stress damage, and apoptosis
	[35]	Mouse	Leydig cells	Testosterone decline	Suppression of GPX1 and upregulation of various stress pathways
	[36]	Mouse	Leydig cells	Increased oxidative, autophagy, and apoptosis	Mitochondrial and endoplasmic reticulum membrane (MAM) dysregulation
(**c**)					
Sertoli cells	Source	Animal/Model	Target Cells	Finding	Mechanism
	[37]	Mouse	Sertoli cells	Decreased tight junction proteins	CHIP-mediated degradation of tight junction proteins by activating IRE1α/XBP1s pathway
	[38]	Mouse	Sertoli cells	Premature senescence	Calcium induced ROS/NF-κB/IL-6,IL-8,TNF-α upregulation
	[39]	Mouse	Sertoli cells	Inflammatory damage	Upregulation of IL-6, IL-10, TGF-β, MCP-1, and TNF-α
	[36]	Mouse	Sertoli cells	Increased oxidative, autophagy, and apoptosis	Mitochondrial and endoplasmic reticulum membrane (MAM) dysregulation

### 2.4. Sertoli Cells

Sertoli cells, which line the seminiferous tubules of the testes, play a foundational role in male reproductive physiology by providing essential structural and functional support for spermatogenesis. These cells establish the blood–testis barrier (BTB), a physical separation between germ cells in the seminiferous tubules and the circulatory system. The BTB shields sperm from potential immune responses—which might otherwise target sperm as foreign cells—and creates a specialized microenvironment necessary for sperm maturation. This barrier is composed of tight junctions, gap junctions, and desmosomes that connect adjacent Sertoli cells. 

MNPs, particularly polystyrene microplastics (PS-MPs), have been shown to disrupt the BTB by degrading tight junction proteins such as occludin, claudin-11, and ZO-2 through the activation of the IRE1α/XBP1s/CHIP pathway [37] in mouse model Sertoli cells. This degradation compromises the barrier integrity, allowing toxins to reach germ cells and enabling interactions between autoantigens and sperm, thereby disrupting the microenvironment required for spermatogenesis [37]. Similar to their impact on Leydig cells, PS-MPs also altered calcium signaling and lipid biosynthesis within Sertoli cells by disrupting mitochondrial-associated endoplasmic reticulum membranes (MAMs), triggering apoptosis and autophagy, which further destabilize the BTB [36] in mouse models. Mitochondrial dysfunction is another critical consequence of PS-MP exposure in Sertoli cells. PS-MPs were found to disrupt the mitochondrial membrane potential (MMP)—the electrical gradient across the mitochondrial membrane—leading to impaired ATP production in a mouse model [38]. A more detrimental outcome of the above disruption is the cascade of opening mitochondrial permeability transition pores (mPTPs), which occurs in response to excessive mitochondrial ROS (mtROS) and then releases ROS and other mitochondrial contents into the cytoplasm [38]. ROS activation then has the potential to impact neighboring mitochondria in a process known as mitochondrial ROS-induced ROS release (RIRR), amplifying cellular damage [38]. Additionally, mitochondrial DNA released via mPTPs activates the cyclic GMP-AMP synthase/stimulator of interferon genes (cGAS-STING) pathway, prompting intracellular calcium release, which is also impacted by MNPs in mouse models [38]. These elevated calcium levels trigger NF-κB activation and the subsequent release of the IL-6, IL-8, and TNF-α—pro-inflammatory markers associated with the senescence-associated secretory phenotype (SASP) [38]. The initiation of inflammatory pathways is a key mechanism underlying Sertoli cell damage. MAP kinase-mediated signaling has been implicated in the accumulation of pro-inflammatory cytokines and chemokines, including IL-6, IL-10, TGF-β, MCP-1, and TNF-α [39], which have been shown to inversely correlate with the number of viable sperm in in vivo mouse studies. Notably, the sperm count reduction was more strongly associated with the duration of MNP exposure [39], although no statistically significant relationship between the MNP dose and sperm count was observed in mice [39]. These data suggest that even low doses of MNPs may negatively affect fertility over prolonged periods; however, this has yet to be directly demonstrated in humans.

Studies from animal and in vitro models collectively suggest that MNPs pose a risk to the structural and functional integrity of Sertoli cells, with potential consequences for male fertility (see Table 1c). The ability of MNPs to cause damage even at relatively low concentrations over extended exposure periods underscores the importance of future research into the long-term implications of MNP toxicity in humans.

### 2.5. Endocrine Disrupting Chemicals as Vectors in MNPs

MNPs act as vectors for endocrine-disrupting chemicals (EDCs), which can adversely affect human reproductive health. EDCs are defined as exogenous chemicals that interfere with hormone action by altering endogenous hormone levels, mimicking natural hormones, or by disrupting hormone synthesis [40]. Common EDCs associated with MNP debris include bisphenol A, phthalates, and polybrominated diphenyl ethers [6]. However, the full impacts of these agents remain difficult to discern due to their latent effects and delayed manifestations [40]. MNPs exacerbate this issue by acting as “molecular sponges”, concentrating EDCs at levels higher than those typically found in the environment [6]. Studies have demonstrated that MNPs can adsorb EDCs such as bisphenol A and phthalates, facilitating their accumulation in biological tissues and intensifying endocrine disruption through synergistic mechanisms [41,42]. Co-exposure models reveal amplified oxidative stress, hormonal imbalance, and reproductive toxicity compared with individual exposures [43]. By intensifying the bioavailability and persistence of EDCs, MNPs transform transient exposures into sustained endocrine challenges, setting the stage for cumulative reproductive dysfunction.

EDCs work through various mechanisms, such as receptor binding, hormone synthesis disruption, and epigenetic reprogramming, which are central to the reproductive toxicity observed. Previous studies exploring this relationship found that BPA functions as a xenoestrogen, binding to estrogen receptors and disrupting the feedback regulation of the hypothalamic pituitary gonadal (HPG) axis [44]. Phthalates impair Leydig cell function and reduce testosterone synthesis, leading to testicular dysgenesis and impaired spermatogenesis [45]. Other common MNP-associated EDCs, such as atrazine and vinclozolin, exhibit anti-androgenic activity and have been shown to induce epigenetic modifications in germ cells, contributing to transgenerational reproductive effects [46,47]. As such, EDC exposure has been linked to disruptions in gametogenesis, gamete quality, pregnancy outcomes, pre- and postnatal development, embryo viability, reproductive tissue integrity, fertility rates, and menstrual cycle regulation—ultimately impairing fertility in both males and females [48] in animal and human studies. In human male studies compiled in WHO manuals from 1999 to 2010, the toxic effects of these compounds included reduced sperm concentration, volume, and motility, as well as other sperm quality parameters [49]. Additionally, studies have linked EDC exposure—including pH-altering compounds—to epigenetic modifications in parental gametes without direct DNA mutation, as well as to germline and embryonic cell mutations that contribute to transgenerational inheritance and altered hormone regulation in mouse pluripotent cells [50]. Research in fish (zebrafish) and mouse models demonstrated that exposure to EDCs such as phthalates, PFAS, DEHP, and mixed EDCs disrupts endocrine function, spermatogenesis, and gonadal development [51,52]. The EDC effects are activated through the mechanisms explored earlier and include increased oxidative stress, DNA damage, testicular transcriptional alterations, and interference with the HPG axis [52]. 

Collectively, these studies suggest that in addition to direct endocrine disruption, MNP–EDC co-exposure in model organisms impairs spermatogenesis and gonadal development through multiple mechanisms, including oxidative stress, DNA damage, and epigenetic alterations (see Table 2 and Figure 3). Future research should investigate whether MNPs and EDC contaminants exert synergistic or antagonistic effects on male reproductive pathways through co-exposure and direct human studies.

## 3. Discussion

Plastics have contributed substantially to advancements across diverse domains of human life; however, emerging evidence indicates that their degradation into MNPs, together with the release of EDCs, may pose significant risks—particularly to male reproductive function. While in vitro and animal models provide a substantial body of data demonstrating the negative health impacts of MNP exposure, direct evidence in humans remains limited. Nevertheless, accumulating findings in model systems raise growing concerns regarding the adverse effects of MNPs on ecological systems, human health, and economic stability.

The research synthesized in this manuscript highlights several recent discoveries regarding the deleterious effects of MNPs on human testicular cells—including germ cells, Leydig cells, and Sertoli cells—and their implications for subfertility. Controlled in vitro and animal models have been instrumental in elucidating biological mechanisms of testicular toxicity. These mechanisms include degradation of the blood–testis barrier, disruption of signaling pathways essential for spermatogenesis and hormone production, induction of oxidative stress and inflammation, structural damage to testicular tissue, DNA fragmentation, and diminished cellular functionality. Functional impairments in sperm capacitation, motility, and metabolic enzyme activity further suggest that MNPs compromise both the structural integrity and functional viability of sperm. In Leydig cells, testosterone synthesis is reduced through mitochondrial dysfunction, ER stress, and transcriptional repression of the LHR/cAMP/PKA/StAR axis. Sertoli cells, which maintain the immune-privileged environment of the seminiferous tubules, are similarly affected by tight junction degradation, mitochondrial ROS release, and inflammatory cytokine accumulation—further destabilizing the microenvironment required for sperm maturation. Importantly, MNPs act as vectors for EDCs, amplifying reproductive toxicity through synergistic mechanisms.

Despite compelling evidence, a major gap in knowledge lies in the direct extrapolation of these findings to human physiology. Unlike controlled in vitro conditions, experimental exposure studies are not feasible in humans due to ethical constraints, leaving most data derived from model systems. Recent studies, however, have confirmed the presence of MNPs in human testis and semen of varying and irregular shapes with particle sizes ranging from 0.72 to 287 μm [53,54,55]. These particles include polypropylene (PP), polyethylene (PE), polyethylene terephthalate (PET), polystyrene (PS), polyvinyl chloride (PVC), polycarbonate (PC), among others [53,54,55]. Although the direct mechanisms of toxicity in the above studies remain unexplored, these findings underscore the urgent need for further investigation into how MNPs affect male fertility.

Future research must prioritize chronic, low-dose exposures that reflect real-world, longitudinal human exposure. While animal models often employ high-dose gavage or direct injection, environmentally relevant exposures are typically lower, ranging from 0.01–10 mg/kg body weight/day in model studies. Recent evidence demonstrates that chronic low-level exposure allows MNPs to infiltrate multiple organs, including reproductive tissues, and induces health effects through long-term inflammation, oxidative stress, and endocrine disruption—changes that are progressive and potentially irreversible [56,57,58,59]. Likewise, interspecies differences in metabolism, physiology, and testicular biology further limit the direct translation of animal findings to human outcomes [60].

Finally, prenatal and early life exposure to MNPs has been shown to impair future fertility in animal models, with persistent effects on testicular structure, hormone regulation, and sperm function. These findings suggest that developmental exposure—whether during gestation or in prepubertal stages—may lead to long-term reproductive toxicity, underscoring the need for human-relevant models and longitudinal studies [61,62].

Collectively, these data highlight the importance of developing ethically sound, human-relevant models. Such efforts are essential to bridge experimental insights with clinical relevance and translate findings into actionable strategies for mitigating reproductive risks associated with MNP exposure.

## 4. Materials and Methods

This literature review was conducted using in accordance with the Scale for the Assessment of Narrative Review Articles (SANRA) guidelines to ensure rigor and reporting quality. A through literature review was conducted using the PubMed, Google Scholar, Scopus, and Web of Sciences databases by all authors of this manuscript, focusing on broad and current relevant studies on the topic. Our references for new findings facilitated our inclusion of total of twenty-one peer-reviewed articles, published in English within the last four years, between January 2019 and October 2025. Search key terms included combinations of the following: microplastics, nanoplastics, male reproductive system, germ cells, Sertoli cells, Leydig cells, and EDC. Priority was given to research relevant to the topic that investigated the cellular-level impacts on germ cells, Leydig cells, and Sertoli cells, incorporating data from animal (specifically in vivo models) and some human studies.

Inclusion criteria: Original studies (both in vivo and in vitro) selected for their direct relevance to the study topic, studies reporting cellular, molecular, or systemic effects (including structural damage, oxidative stress and endocrine interference) of MNPs and/or EDCs on male reproductive tissues or hormones, and studies published in English.

Exclusion criteria: Studies unrelated to the above criteria, studies without primary data and studies that lacked a clear methodology (defined as insufficient detail regarding study design, participant selection, data collection, or analysis procedures), or if they lacked specific quantitative findings or identifiable qualitative themes that support the study’s conclusions.

Data extraction and organization: Titles and abstracts were screened for relevance. Full-text articles meeting the inclusion criteria were reviewed. Data were extracted and categorized regarding study model (animal, human, in vivo), cell type affected (germ, Sertoli, Leydig) and the nature of disruption (e.g., morphological changes, hormonal dysregulation). This thematic classification enabled a clearer synthesis of emerging patterns and mechanistic insights across study models. Results were organized and summary tables were constructed to summarize the findings.

Limitations: This review predominantly draws upon preclinical evidence drawn from model organism studies. While relevant human data are incorporated where available, findings are interpreted with caution to avoid over-extrapolation. This review also concludes by identifying key gaps in current knowledge and outlining priorities for future research. As such, limitations inherent to the existing evidence base—such as variability in study design and translational relevance—are acknowledged and critically discussed.

## 5. Conclusions and Future Directions

While substantial advances have been made in elucidating the reproductive toxicity of micro- and nanoplastics, critical knowledge gaps persist, particularly regarding human-relevant exposures and outcomes. Future research must prioritize the development and application of human-centric models to capture both acute and chronic effects, incorporating realistic exposure scenarios that reflect environmental complexity. Expanding epidemiological studies across diverse populations and variations (lifestyle, dietary habits, and environmental factors) and refining simulation models will be essential to translate mechanistic insights into effective public health strategies. Ultimately, advancing these approaches will enable a comprehensive understanding of MNPs’ reproductive impact and inform targeted interventions to mitigate associated risks.

## Figures and Tables

**Figure 1 ijms-26-11457-f001:**
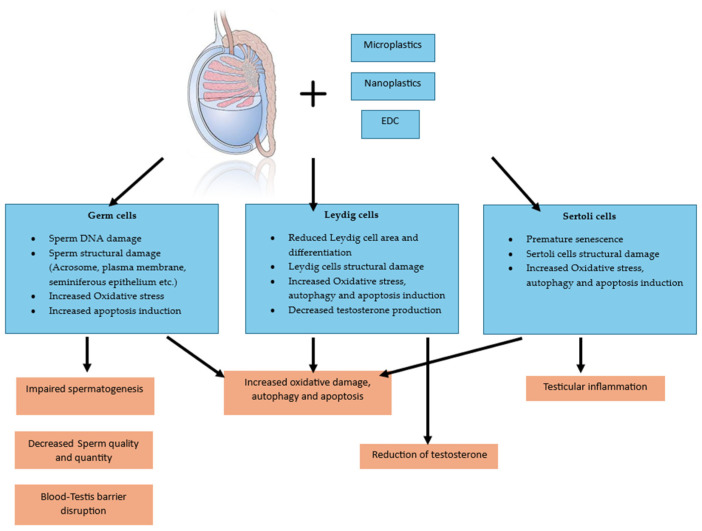
Summary of the Cellular and Hormonal Disruption in the Testis Caused by Microplastics, Nanoplastics, and Endocrine Disruptors in analyzed in vitro and in vivo models.

**Figure 2 ijms-26-11457-f002:**
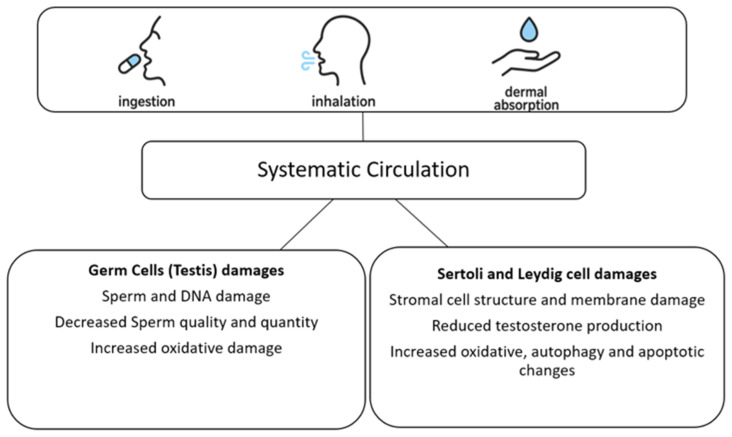
Graphical algorithm of micro- and neoplastic (MNP) effects on testis–germ cells and stromal (Sertoli and Leydig) cells.

**Figure 3 ijms-26-11457-f003:**
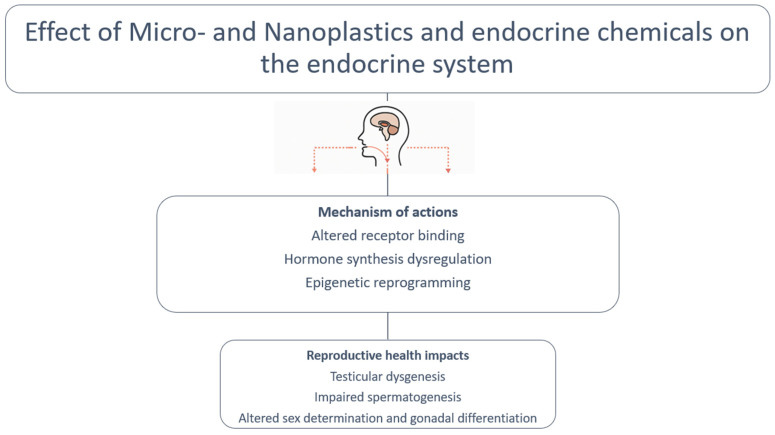
The effects of micro- and nanoplastics on the endocrine system.

**Table 2 ijms-26-11457-t002:** Summary of the MNP-endocrine disrupting chemicals’ co-exposure effects on the male reproductive system. These data suggest MNP–EDC co-exposure in model organisms impairs spermatogenesis and gonadal development through multiple mechanisms, including oxidative stress, DNA damage, and epigenetic alterations.

Endocrine disrupting effects	Source	EDC Studied	Animal/Model	Target Cells	Finding	Mechanism
	[50]		Mouse Pluripotent	Sertoli, Leydig and germ cells	Dysregulation of normal gene/epigenetic functioning	Disruption of the chromatin structure that usually corrects the epigenetic mutation
	[51]	Phthalates, PFAS, DEHP, Mixed EDCs from paper mills	Zebrafish	Germ (Spermatogenic) cells	Altered sex determination and gonadal differentiation	Methylation of gonadal developmental pathways such as PPAR, ERR, ROS, PI3K, AHR, and RA.
	[52]	Phthalates	Male mouse	All testicular (mostly Sertoli)	Increased negative alterations in sperm physiology and spermatogenesis	Increased oxidative stress and alternation in the testicular transcriptomic gene

## Data Availability

No new data were created or analyzed in this study. Data sharing is not applicable to this article.

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
