# Peer review of "The Threat of Micro-/Nanoplastics to Male Fertility: A Review of the Data and the Importance of Future Research"

_ijms, 2025, doi:10.3390/ijms262311457_

Round 1

Reviewer 1 Report

Comments and Suggestions for Authors

The study integrates 21 studies published between January 2019 and March 2025, focusing on the cellular-level impacts of MNPs and EDCs on germ cells, stromal cells, and support cells. It reveals various testicular toxic mechanisms, including blood-testis barrier degradation, disruption of key signaling pathways involved in spermatogenesis and hormone synthesis, induction of oxidative stress and inflammation, as well as structural and genetic damage to testicular tissue. The overall structure of the article is coherent, the argumentation is robust, and data support is sufficient. However, significant revisions are required before potential publication.

  1. In line 13 of the manuscript, "MNPs" is explained, but the term "MP" in line 125 and "MNP" in line 132 are not explained. It is recommended to provide appropriate clarifications for these terms.
  1. In line 140, the comma before "and" is redundant and should be removed. Additionally, the period before the citation symbol in line 326 should be deleted. In line 160, a period should be added after the word "models."
  1. The use of "proliferation" in line 173 is incorrect; it is suggested to replace it with "upregulation." In line 178, the word "lead" is used incorrectly; it should be changed to "led."
  1. In line 296, the use of "Where" is somewhat inappropriate; it is recommended to change it to "In the model systems highlighted here." In line 188, the article "a" should be deleted, and the abbreviation "BTB" in line 197, which appears for the first time, should be explained.
  1. The article only conducted a literature search in the PubMed and Google Scholar databases. To enhance the article's persuasiveness, it is recommended to include additional databases, such as Web of Science.
  1. The use of "its" in line 324 is incorrect, and it is suggested to replace it with "their."
  2. In the introduction, the article does not address whether any studies have reported the presence of microplastics and EDCs in the human reproductive system. If such reports exist, what are the compositions, morphologies, and particle sizes of the microplastics?
  1. Although the article mentions the necessity of "chronic, low-dose exposure," it may require a more detailed explanation of the potential mechanisms through which chronic exposure could affect reproductive health. For instance, would small changes in hormone levels due to prolonged exposure accumulate into significant physiological effects?
  1. The definition of "low-dose exposure" could be further discussed, including which studies have already explored long-term low-dose exposure. Additionally, it would be helpful to examine how exposure levels and durations in these studies affect the experimental outcomes.
  1. Microplastics themselves may be environmental pollutants, and as carriers of EDCs, they could potentially exacerbate the endocrine-disrupting effects of these chemicals. It is recommended that the article further explore the synergistic effects of microplastics in combination with other EDCs, such as whether microplastics can accelerate the accumulation of EDCs in the body or intensify their toxic effects.
  1. In the conclusion, specific directions for future research could be proposed, such as: increasing population-based epidemiological studies to better assess the long-term effects of microplastics and EDCs; enhancing simulation studies of different environmental exposure scenarios to explore variations in exposure due to lifestyle, dietary habits, and environmental factors.
  1. The article mentions that endocrine-disrupting chemicals (EDCs) may have an impact on male reproductive health, but it does not delve into which specific types of EDCs are most representative or their mechanisms of action. A more detailed classification and analysis of the mechanisms of common EDCs, such as bisphenol A, phthalates, pesticides, etc., could be discussed.

Author Response

1. In line 13 of the manuscript, "MNPs" is explained, but the term "MP" in line 125 and "MNP" in line 132 are not explained. It is recommended to provide appropriate clarifications for these terms.

Response 1: Thank for the recommendations and in the revised manuscript we have made the suggested changes

2. In line 140, the comma before "and" is redundant and should be removed. Additionally, the period before the citation symbol in line 326 should be deleted. In line 160, a period should be added after the word "models."

Response 2: Thank for the recommendations and in the revised manuscript we have made the suggested changes

3. The use of "proliferation" in line 173 is incorrect; it is suggested to replace it with "upregulation." In line 178, the word "lead" is used incorrectly; it should be changed to "led."

Response 3: We have substituted those terms (Line 173 proliferation to upregulation and Line 178 lead to led) as indicated.

4. In line 296, the use of "Where" is somewhat inappropriate; it is recommended to change it to "In the model systems highlighted here." In line 188, the article "a" should be deleted, and the abbreviation "BTB" in line 197, which appears for the first time, should be explained.

Response 4: We have removed the “where” in line 296, deleted “a” in line 188 and expanded on the abbreviation for BTB as blood-testis barrier in first use.

5. The article only conducted a literature search in the PubMed and Google Scholar databases. To enhance the article's persuasiveness, it is recommended to include additional databases, such as Web of Science.

Response 5: We have modified the materials and methods sections to include PubMed, Google Scholar and other smaller databases we used.

6. The use of "its" in line 324 is incorrect, and it is suggested to replace it with "their."

Response 6: We couldn’t find the use of the word “its” on line 324 but on line 342. We agree with the use in the context and agreeably change to the suggested word “their”.

7. In the introduction, the article does not address whether any studies have reported the presence of microplastics and EDCs in the human reproductive system. If such reports exist, what are the compositions, morphologies, and particle sizes of the microplastics?

Response 7: Thank you for pointing this out. We have now included a study in understanding the mechanism of MNPs induced germ cell toxicity done on human spermatozoa, and have included details as to the compositions, particle size, and concentrations of MNPs found in humans. Likewise, we have included few similar studies in discussion that have reported the presence of MNPs in human male reproductive system including in the semen and testis and its appropriate available details.

8. Although the article mentions the necessity of "chronic, low-dose exposure," it may require a more detailed explanation of the potential mechanisms through which chronic exposure could affect reproductive health. For instance, would small changes in hormone levels due to prolonged exposure accumulate into significant physiological effects?

9. The definition of "low-dose exposure" could be further discussed, including which studies have already explored long-term low-dose exposure. Additionally, it would be helpful to examine how exposure levels and durations in these studies affect the experimental outcomes.

Response 8 and 9: The manuscript brings forward the necessity of “chronic low dose exposure” in the concluding paragraph to highlight the necessity of future studies to explore not only the how exposure occurs but also how duration of exposure may impact fertility. As far we know, there are only a few studies which have explored this dynamic and mechanism such as long-term inflammation, oxidative stress and endocrine damage that is progressive and potentially irreversible in humans, but due to the fact no human clinical trial can be conducted in humans, this information remains speculative.

10. Microplastics themselves may be environmental pollutants, and as carriers of EDCs, they could potentially exacerbate the endocrine-disrupting effects of these chemicals. It is recommended that the article further explore the synergistic effects of microplastics in combination with other EDCs, such as whether microplastics can accelerate the accumulation of EDCs in the body or intensify their toxic effects.

Response 10: We agree with this suggestion. As such we have edited and added new studies to our manuscript that analyze the “synergistic effects of microplastics in combination with other EDCs…” in lines 250-256.

11. In the conclusion, specific directions for future research could be proposed, such as: increasing population-based epidemiological studies to better assess the long-term effects of microplastics and EDCs; enhancing simulation studies of different environmental exposure scenarios to explore variations in exposure due to lifestyle, dietary habits, and environmental factors.

Response 11: Per reviewer suggestion, we have modified the conclusion section and included the suggested discussion points for future research in lines 399-401.

12. The article mentions that endocrine-disrupting chemicals (EDCs) may have an impact on male reproductive health, but it does not delve into which specific types of EDCs are most representative or their mechanisms of action. A more detailed classification and analysis of the mechanisms of common EDCs, such as bisphenol A, phthalates, pesticides, etc., could be discussed.

Response 12: Per reviewer’s suggestion, we now discuss EDCs mechanisms of action—such as xenoestrogenic receptor binding, androgen synthesis disruption, and epigenetic reprogramming—and their documented effects on male reproductive health in lines 261-269.

Reviewer 2 Report

Comments and Suggestions for Authors

The review is highly relevant to the field of reproductive biology, particularly male fertility.
The reviewer finds it to be comprehensive in its description of the effects of micro- and nanoplastics on male fertility, including their impact on the various testicular cell types and the endocrine system.

The reviewer would like to suggest that the following subjects be addressed:

  1. describing the effects of micro- and nanoplastics not only on adult fertility but also on the future fertility of prepubertal boys and of animals exposed to these materials, either directly or during pregnancy.

  2. Add a figure (or chart) illustrating the effects of micro- and nanoplastics on the endocrine system.
  3. Provide specific examples of micro- and nanoplastics (and their sources) that have been shown to affect the endocrine system, as well as those that specifically impact individual testicular cell types (somatic or germ cells).

Author Response

1. Describing the effects of micro- and nanoplastics not only on adult fertility but also on the future fertility of prepubertal boys and of animals exposed to these materials, either directly or during pregnancy.

Response 1: We appreciate the reviewer’s thoughtful suggestion. While our current manuscript focuses primarily on adult reproductive toxicity, expanding the scope to include prepubertal and developmental exposures would require a distinct framework and dataset beyond the present analysis. However, we agree that this is a critical and emerging area of investigation and emphasized in lines 355-359 with recent studies.

2. Add a figure (or chart) illustrating the effects of micro- and nanoplastics on the endocrine system.

Response 2: We added a figure 2 that explores the EDC effects on the male reproductive system as mentioned in section 2.5.

3. Provide specific examples of micro- and nanoplastics (and their sources) that have been shown to affect the endocrine system, as well as those that specifically impact individual testicular cell types (somatic or germ cells).

Response 3: Most of the studies referenced in our manuscript are based on in vitro and in vivo experimental models, which typically do not specify the environmental sources of micro- and nanoplastics (MNPs) or endocrine-disrupting chemicals (EDCs). However, we have addressed the reviewer’s concern by emphasizing the documented impacts and mechanisms of action of representative MNPs and EDCs—such as bisphenol A, phthalates, and pesticides—on testicular function in line 261-269 and on individual testicular cell types (somatic or germ cells) if available.

Reviewer 3 Report

Comments and Suggestions for Authors

The review titled "Micro- and Nanoplastics as Threats to Human Male Fertility: A Cellular Review of Reproductive Toxicity and Endocrine Disruption" by Shawn et al. collects the latest studies regarding the activity of micro- and nanoplastics and the probable associated EDCs on the male gonad and spermatogenesis. The review is well organized and requires minor changes.

In my opinion, the title does not reflect the content of the review well, as the articles consulted primarily use model organisms, particularly mose and rat.

"In vitro" and "in vivo" should be italicized.

Eliminate spaces between paragraphs within the same chapter.

Check for spaces and typos.

Author Response

1. In my opinion, the title does not reflect the content of the review well, as the articles consulted primarily use model organisms, particularly mouse and rat.

Response 1: Thank you for the suggestion, we have now changed the title of the article to “Micro- and Nanoplastics as Threats to Male Fertility: A Cellular Review of data and the importance of future research

2. "In vitro" and "in vivo" should be italicized.

Response 2: Thank you and we have made the changes accordingly.

3. Eliminate spaces between paragraphs within the same chapter.

Response 3: Thank you and we have made the changes accordingly.

Round 2

Reviewer 1 Report

Comments and Suggestions for Authors

In the revised version of the manuscript the authors carefully responded to all the suggestions made by the reviewer, greatly improving the quality of the paper. In my opinion,  the manuscript may be accepted for publication.

Author Response

Thank you for the suggestion and help

Reviewer 3 Report

Comments and Suggestions for Authors

I would remove "cellular" from the title

Author Response

I would remove "cellular" from the title

Response - Thank you for the suggestion, we have incorporated the change